# The Chimeric Adenovirus (Ad5/35) Expressing Engineered Spike Protein Confers Immunity against SARS-CoV-2 in Mice and Non-Human Primates

**DOI:** 10.3390/vaccines10050712

**Published:** 2022-04-30

**Authors:** Seung-Phil Shin, Kwang-Soo Shin, Jeong-Mi Lee, In-Kyung Jung, Jimo Koo, Seung-Woo Lee, Seowoo Park, Jieun Shin, Myunghwan Park, Bongju Park, Hanseul Oh, Bon-Sang Koo, Jungjoo Hong, Choong-Min Ryu, Jae-Ouk Kim, Taegwon Oh, Chang-Yuil Kang

**Affiliations:** 1Cellid Co., Ltd., Seoul 08826, Korea; apollossp@gmail.com (S.-P.S.); ksshin@cellid.co.kr (K.-S.S.); cipe@naver.com (J.-M.L.); ikjung@cellid.co.kr (I.-K.J.); koo-chj@hanmail.net (J.K.); zhizhu55@gmail.com (S.-W.L.); swpark@cellid.co.kr (S.P.); jeshin@cellid.co.kr (J.S.); angelouss@naver.com (M.P.); bjpark@cellid.co.kr (B.P.); taegwon1132@gmail.com (T.O.); 2National Primate Research Center, Korea Research Institute of Bioscience and Biotechnology, Cheongju 28116, Korea; seul3198@kribb.re.kr (H.O.); porco9@kribb.re.kr (B.-S.K.); hong75@kribb.re.kr (J.H.); 3Infectious Disease Research Center, Korea Research Institute of Bioscience and Biotechnology, Daejeon 34141, Korea; cmryu@kribb.re.kr; 4Science Unit, International Vaccine Institute, Seoul 08826, Korea; jokim@ivi.int; 5Laboratory of Immunology, Research Institute of Pharmaceutical Science, College of Pharmacy, Seoul National University, Seoul 08826, Korea

**Keywords:** SARS-CoV-2, variants, COVID-19 vaccine, chimeric adenovirus-vectored vaccine, GS linker, neutralizing activity, T_h_1 immune responses

## Abstract

Several COVID-19 platforms have been licensed across the world thus far, but vaccine platform research that can lead to effective antigen delivery is still ongoing. Here, we constructed AdCLD-CoV19 that could modulate humoral immunity by harboring SARS-CoV-2 antigens onto a chimeric adenovirus 5/35 platform that was effective in cellular immunity. By replacing the S1/S2 furin cleavage sequence of the SARS-CoV-2 Spike (S) protein mounted on AdCLD-CoV19 with the linker sequence, high antigen expression was confirmed in various cell lines. The high levels of antigen expression contributed to antigen-specific antibody activity in mice and non-human primates (NHPs) with a single vaccination of AdCLD-CoV19. Furthermore, the adenovirus-induced T_h_1 immune response was specifically raised for the S protein, and these immune responses protected the NHP against live viruses. While AdCLD-CoV19 maintained neutralizing antibody activity against various SARS-CoV-2 variants, it was reduced to single vaccination for β and ο variants, and the reduced neutralizing antibody activity was restored with booster shots. Hence, AdCLD-CoV19 can prevent SARS-CoV-2 with a single vaccination, and the new vaccine administration strategy that responds to various variants can maintain the efficacy of the vaccine.

## 1. Introduction

SARS-CoV-2 is an enveloped single-stranded positive-sense RNA virus that belongs to the family Coronaviridae [1]. COVID-19, caused by SARS-CoV-2, is characterized by clinical sequelae such as fever, shortness of breath, cough, loss of smell, and loss of taste following a latent period ranging from 2 to 14 days, or maybe asymptomatic [2,3]. Although the mortality reported for COVID-19 accounts for only a small proportion compared with the total number of people infected, its rapid contagion has caused millions of deaths worldwide [4,5,6,7]. Moreover, despite the efforts of many research groups to accelerate the development of vaccines and therapeutics, the variant forms of SARS-CoV-2 with unstable genomes [6], including ‘Variant of concern (VOC)’ [8], have raised more concern about their transmissibility and antigenicity [9,10,11]. These VOCs, in particular, are acquiring faster infectivity as a result of an evolutionary process in which their spike proteins are modified to increase receptor tropism [12].

The spike (S) protein of SARS-CoV-2, the component that defines transmissibility, is cleaved into the S1 and S2 subunits by host cell proteases [13,14]. The S2 subunit containing the fusion peptide promotes membrane fusion and viral entry, while the S1 subunit binds to the ACE2, the receptor on host cells, through the receptor-binding domain (RBD) [14,15]. In addition, the S protein mutations of the novel variants exist in the RBD motif, which leads to increased infectivity over wild-type [15,16]. All the variant viruses harbor multiple mutations in the S protein, which is a crucial determinant of viral infectivity and the region where most SARS-CoV-2 mutations occur [17]. Representatively the Delta (B.1.617.2, accession number: MZ208926.1) or Omicron (B.1.1.529, accession number: OL869974.1) variants underwent significant conformational changes in both the N-terminal domain (NTD) and the RBD of the spike protein, allowing effective membrane fusion at low levels of the cell receptor angiotensin-converting enzyme 2 (ACE2), which explains their improved infectivity [12,18]. Therefore, the effectiveness of a COVID-19 vaccine depends on proper cooperation between B cells and T cells capable of blocking virus-host cell contact and killing infected cells.

Among various vaccines, recombinant adenoviral vector platforms are specialized in eliciting potent CD8^+^ T cell responses [19]. The chimeric adenovirus vector (Ad5/35), in which the fiber knob is replaced with that of type 35, activates antigen-presenting cells, such as monocytes, via secretion of inflammatory cytokines from the transduced cells, and elicits T_h_1-biased immune responses while excluding the adverse effects triggered by the T_h_2 reaction [20]. CD46, a receptor of serotype 35 fiber, enables the Ad5/35-based vaccine to promote CD8^+^ T cell responses due to its frequent expression in antigen-presenting cells [21]. Thus, Ad5/35 adenoviral vectors can ensure high transduction efficiency and induction of T_h_1-biased immunity as a vaccine carrier.

To elevate the humoral immune responses that balance cell-mediated immune responses, it is effective to induce overexpression of the antigen-loaded into the gene delivery system. The serum level and affinity of neutralizing antibodies induced by vaccination are associated with the amount of antigen that naive B cells encounter [22,23]. Therefore, effective COVID-19 vaccine candidates must adopt the structural antigen design for efficient antigen delivery [24]. The strategies of substituting the tissue plasminogen activator (tPA) [25,26] or proline (K986P, V987P) [27,28,29,30] for the spike protein have been applied to various COVID-19 vaccine candidates in an attempt to improve the efficiency of antigen expression.

As a protein engineering technique, polypeptide linkers consisting of glycine and serine (GGGGS)_n_ are used to connect the protein domains and improve the stability of the region [31,32]. Since the insertion of a flexible GS linker, (GGGGS)_n_ for example, can improve the folding and stability of fusion proteins, we aimed to increase the antigenicity of the vaccine by placing the linker between the S1 and S2 domain sequences of SARS-CoV-2.

In summary, we propose the adenovirus-vectored vaccine, AdCLD-CoV19, which is designed to optimize the expression of S protein using a replication-deficient recombinant adenovirus serotype 5/35 and the linker sequence. To test the promising effect of the S protein sequence modification strategy, we assessed the immunogenicity and protective efficacy of AdCLD-CoV19 at various doses compared to the wild-type vaccine that lacks the linker sequence. We also evaluated whether neutralizing antibodies produced by AdCLD-CoV19 could counteract single point mutations observed in novel variants of COVID-19 and the effectiveness of the booster shot strategy. This study will provide evidence for further clinical use of AdCLD-CoV19 and underline the validity of modifying the S protein sequence to develop effective adenovirus-vectored vaccines.

## 2. Results

### 2.1. The Modified S Protein Expression in Various Cell Types

To find an optimal design of the prophylactic vaccine against SARS-CoV-2 that can efficiently induce the expression of S protein in host cells, we constructed E1/E3 deleted replication-deficient recombinant Ad5/35 candidates capable of efficiently expressing the SARS-CoV-2 S protein in transduced cells (Figure 1a, Appendix A). The S protein-coding sequence of each candidate can be easily exposed to immune cells by allowing sufficient amounts of antigen to be released out of the cell through codon optimization and truncation of the transmembrane–cytoplasmic tail. The linker sequence (GGGGS)_1_ of AdCLD-S.GS was designed to induce antigen overexpression in various cells by substituting it for the furin cleavage site between the S1 and S2 domains (Figure 1a). To verify whether the vaccine candidates can expose the host immune system to a sufficient amount of S protein, we evaluated their ability to induce S protein expression in various types of human cell lines, including the monocytic cell line THP-1, epithelial cell line A549, and muscle cell line RD. AdCLD-S.GS most efficiently induced high levels of S protein expression in all tested cell lines compared to AdCLD-S.WT that was used as a control (Figure 1b–d). In addition, AdCLD-S.GS was compared with AdCLD-S.IL, and AdCLD-S.GS3, the linkers Isoleucine–Leucine (IL) and (GGGGS)_3_ were additionally constructed. In addition, AdCLD-S.GS3 showed a satisfactory expression efficiency in each cell line. AdCLD-S.GS elicited higher levels of secretion antigen expression in THP-1 cells than AdCLD-S.GS3 (Appendix A).

These data show that adenovirus-vectored AdCLD-S.GS, named AdCLD-CoV19, displayed consistent transduction efficiency in THP-1, A549, and RD, suggesting its capacity to deliver S protein antigen to the injection site and immune cells in vaccinated hosts.

### 2.2. Immunogenicity of AdCLD-CoV19 in Mice

To test the immunogenicity of AdCLD-CoV19, BALB/c mice were vaccinated with a single intramuscular injection of 1 × 10^9^ VP of AdCLD-CoV19 or AdCLD-S.WT. We examined S protein-specific antibodies in vaccine sera obtained at 2–7 weeks post-vaccination. We found that AdCLD-CoV19-vaccinated mice exhibited significantly higher titers of S protein-specific binding antibodies than those treated with AdCLD-S.WT from 3 weeks post-vaccination (Figure 2a and Appendix A). To evaluate the neutralizing ability of the vaccine-induced antibodies, we performed a SARS-CoV-2 S-protein-expressing pseudotyped lentivirus-based neutralizing assay (Appendix A). Neutralizing titers of AdCLD-CoV19 vaccine sera were higher than those of AdCLD-S.WT vaccine sera on 5 and 7 weeks post vaccination (Figure 2b and Appendix A). A low dose (2 × 10^8^ VP) of AdCLD-CoV19 showed intact antibody production, although there was a slight drop in neutralizing titers at week 4 (Figure 2c,d and Appendix A). In summary, these data suggest that AdCLD-CoV19 induces S protein-specific humoral immune responses.

To determine whether T cell immune responses are accompanied by antibody production after vaccination, we conducted an in vivo CTL analysis for AdCLD-CoV19-vaccinated mice. Two weeks after vaccination, the peptide pools of S protein-loaded target cells were injected into mice, and target cell lysis was measured. There was no specific lysis found in naïve mice; however, peptide-pulsed target cells in vaccinated mice were significantly reduced in their percentages compared to unpulsed cells (Figure 2e). Thus, vaccination with AdCLD-CoV19 could induce CTL responses in an Ag-specific manner. Based on these findings, ELISpot assay and flow cytometry analysis showed that T cells existing vaccine sera could secrete IFN-γ and proliferate well under peptide pool stimulation (Figure 2f,g).

Thereafter, we assessed the T_h_1/T_h_2 cytokine profiles in vaccinated mice. For activated CD8^+^ T cells, the levels of T_h_1 cytokines (IFN-γ and TNF-α) were elevated in vaccinated mice compared with control mice, whereas T_h_2 cytokines (IL-4 and IL-5) were scarcely detected in either group (Figure 2h). Similarly, activated CD4^+^ T cells also showed T_h_1-skewed profiles (Figure 2h). Therefore, a single dose of AdCLD-CoV19 induced S protein-specific binding antibodies, neutralizing antibodies and T_h_1-biased T cell responses in mice.

### 2.3. Immunogenicity of AdCLD-CoV19 in NHPs

To evaluate the immunogenicity of AdCLD-CoV19 in cynomolgus macaques (non-human primates, NHPs), we administered naïve NHPs intramuscularly with 1 × 10^11^ VP of AdCLD-S.WT or AdCLD-CoV19. For both groups, high titers of S protein-specific binding antibodies were observed in all vaccinated monkeys from two weeks post-vaccination, which persisted until week 5 (Figure 3a and Appendix A). While there was no difference in binding antibody titers between the two groups, the serum neutralizing titers of Ad-CLD-CoV19-vaccinated animals were higher than those of AdCLD-S.WT-vaccinated animals, as evidenced by the pseudovirus neutralizing assay (Figure 3b and Appendix A). We observed that AdCLD-CoV19-vaccinated animals maintained high amounts of neutralizing Abs until 5 weeks after a single dose of vaccination, whereas there was a declining trend in neutralizing Ab titers of the sera from AdCLD-S.WT-vaccinated animals (Figure 3b and Appendix A). Similarly, the result of the focus reduction neutralizing test (FRNT) showed that the serum samples collected from AdCLD-CoV19-vaccinated animals 3 weeks after vaccination had higher neutralization activity against live SARS-CoV-2 compared with AdCLD-S.WT-vaccinated animals (Figure 3c and Appendix A). To address the dose-dependent immunogenicity of AdCLD-CoV19, we immunized eight naive NHPs with a single administration of AdCLD-CoV19 at three dose levels: low-dose group 2 × 10^10^ VP (*n* = 3); medium-dose group 4 × 10^9^ VP (n = 3); and high-dose group 1 × 10^11^ VP (n = 2). Serum antibody titers using ELISA showed that medium-dose AdCLD-CoV19 elicited potent humoral responses to a degree equivalent to high-dose, whereas low-dose was less effective in antibody production compared to higher doses (Figure 3d and Appendix A). Although there was no significance except 5 weeks post-vaccination due to the limited animal numbers, the sera from medium-dose and high-dose groups showed a comparable capacity to neutralize the pseudotyped virus, in contrast to the neutralizing activity of low-dose vaccine sera (Figure 3e and Appendix A). These data suggest that medium-dose and high-dose AdCLD-CoV19 exhibited a comparable efficacy profile on humoral immune responses in NHPs.

Thereafter, we investigated T cell responses induced by AdCLD-CoV19 vaccination. We found that IFN-γ-producing antigen-specific T cells were present in the peripheral blood mononuclear cells (PBMCs) of all vaccinated NHPs (Figure 3f). There was no significant difference in the spot-forming units between the medium-dose and high-dose groups, indicating that a medium dose of AdCLD-CoV19 is sufficient to induce optimal T cell responses in NHPs. In summary, we demonstrated that AdCLD-CoV19 facilitates immune responses against SARS-CoV-2 mediated by neutralizing antibodies and antigen-specific T cells in NHPs.

### 2.4. Protective Efficacy of AdCLD-CoV19 in NHPs

To address the protective efficacy of AdCLD-CoV19, we vaccinated naïve cynomolgus macaques with a single medium dose (2 × 10^10^ VP) of AdCLD-CoV19. The Ad5/35 vector carrying the GFP gene was used as a control. We confirmed that all vaccinated NHPs displayed antigen-specific T cell responses and generated S protein-specific Abs and SARS-CoV-2 neutralizing Abs after vaccination (Appendix A). Seven weeks after single-shot vaccination, we challenged vaccinated NHPs with the wild-type SARS-CoV-2 virus via multiple routes as established in a previous study [33]. All Ad5/35-GFP-received NHPs had elevated body temperature one day after the virus challenge; however, AdCLD-CoV19-received NHPs remained within the normal range except for one macaque, which showed a slight increase in body temperature after-challenge (Figure 4a). To verify whether respiratory virus shedding can be reduced by vaccination, we obtained nasal and throat swab samples from vaccinated macaques. The viral loads were measured by qRT-PCR. Although viral load in the nasal swabs was not significantly suppressed by vaccination, we observed a significant decrease in viral load in throat swabs from vaccinated macaques, resulting in undetectable levels three days after the virus challenge (Figure 4b). Moreover, the amount of infectious virus quantified by the median tissue culture infectious dose (TCID_50_) reached undetectable levels in nasal swabs and throat swabs two days after the virus challenge (Figure 4c). These data suggest that AdCLD-CoV19 can play a role in preventing SARS-CoV-2 by promoting viral clearance from the upper respiratory tract.

To assess the effects of AdCLD-CoV19 against pulmonary involvement by SARS-CoV-2, we collected lung specimens from the macaques that were euthanized three days after the challenge. qRT-PCR analysis revealed that viral loads in the bronchus and left and right lobes of lung tissue samples were significantly reduced in vaccinated NHPs compared to control NHPs (Figure 4d,e). Invasion of the lungs by infectious SARS-CoV-2 was completely disrupted by AdCLD-CoV19 (Figure 4f). In summary, a single shot of AdCLD-CoV19 confers protection against SARS-CoV-2 in NHPs.

### 2.5. The Neutralizing Effects of AdCLD-CoV19 against SARS-CoV-2 Variants

To evaluate the neutralizing activity of AdCLD-CoV19 against B.1.1.7, B.1.351, P.1, B.1.617.2, and B.1.1.529 variants (VOCs) we constructed a pseudotyped lentivirus harboring the mutated S protein. In addition, we assessed pseudotyped lentiviruses that had significant mutations that were widespread in each variant (Figure 5a).

The pseudotyped lentiviral variants reacted with sera from 4, 5, or 11 weeks of vaccination via a single intramuscular injection of either medium (M; 2 × 10^10^ VP) or high (H; 1 × 10^11^ VP) doses of AdCLD-CoV19. Although we were limited in deriving significant statistics because only a small number of NHPs (2–3 monkeys/group) were sacrificed considering animal ethics in the experiment, we could confirm the tendency. The D614G and N501Y mutations, which are common to various variants, did not reduce the vaccine effect against AdCLD-CoV19 (Figure 5b,c); however, the neutralizing activity of the B.1.1.7 variant was slightly decreased in the medium- and high-dose groups compared to the Wuhan-Hu-1 wild-type group (GMT: M_WT vs. Variant: 330.4 vs. 223.9, 1.5-fold; H_WT vs. Variant: 519 vs. 406.8, 1.3-fold) (Figure 5d).

The neutralizing antibody titer for the B.1.351 variant, on the other hand, was significantly more reduced than the wild type (GMT: M_WT vs. Variant: 216 vs. 33.7, 6.4-fold; H_WT vs. Variant: 315.6 vs. 90.8, 3.5-fold), and the key mutations of B.1.351 (partial) and K417N/E484K/N501Y followed a similar tendency (Figure 5e–g). Furthermore, the neutralizing antibody activity against B.1.1.529, which also contains the 417/484/501 site point mutation, decreased in a similar pattern to that of B.1.351 (GMT: M_WT vs. Variant: 279.6 vs. 29.4, 9.5-fold; H_WT vs. Variant: 650.7 vs. 63.7, 10.2-fold) (Figure 5l). Interestingly, the P.1 variant, which shares important mutations with B.1.351 and B.1.1.529, restored neutralizing activity compared to B.1.351 (GMT: M_WT vs. Variant: 278.5 vs. 225.2, 1.2-fold; H_WT vs. Variant: 521.1 vs. 338.9, 1.5-fold) (Figure 5h).

In the B.1.617.2 variant, the neutralizing activity was decreased compared to that of the wild type at the medium dose; however, it recovered significantly at the high dose (GMT: M_WT vs. Variant: 231.6 vs. 66.9, 3.5-fold; H_WT vs. Variant: 485.2 vs. 533.7, 0.9-fold) (Figure 5k). This trend was also confirmed in the L452R/T478/D614G variant, which has a common mutation with B.1.617.2 (GMT: M_WT vs. Variant: 171.5 vs. 116.1, 1.5-fold; H_WT vs. Variant: 401.2 vs. 484.2, 0.8-fold) (Figure 5i). The neutralizing antibody activity was decreased in common among the variants containing the S protein amino acid sequence 484 mutation (Figure 5e–h,j,l).

Taken together, these results demonstrate that AdCLd-CoV19 elicited effective neutralizing antibodies after a single dose, but this reduced for several variants, including B.1.351 and B.1.1.529. This suggests that the AdCLD-CoV19 injection strategy should be improved.

### 2.6. Booster Injection of AdCLD-CoV19-1 against SARS-CoV-2 Variants

To determine whether the booster injection of AdCLD-CoV19 could restore the reduced neutralizing antibody activity by ‘VOC’, we analyzed the pseudovirus neutralization test in NHPs.

Before the experiment, we developed AdCLD-CoV19-1, a rearranged version of adenovirus E4orf6, to minimize the incidence of replication-competent adenovirus (RCA), a long-standing issue in AdCLD-CoV19 large scale production. The antigen expression system on AdCLD-CoV19 and AdCLD-CoV19-1 is identical, and it has been verified that there is no difference in antigen expression level and the function of humoral/cellular immunity (Appendix A).

To enhance the resolution of the booster injection effects, we delivered the same dose (2 × 10^10^ VP) of booster injection in the 9 weeks after intramuscular injection of NHPs once at the medium dosage (2 × 10^10^ VP) that was susceptible to ‘VOC’. NHPs reimmunized by booster injection acquired effective neutralizing antibody activity in ‘VOC’ (GMT: WT_9w vs. 11w: 114.7 vs. 3537, 30.8-fold; B.1.351_9w vs. 11w: 32.2 vs. 1088, 33.8-fold; P.1_9w vs. 11w: 307.3 vs. 3914, 12.7-fold; B.1.617.2_9w vs. 11w: 34.9 vs. 863.1, 24.7-fold; B.1.1.529_9w vs. 11w: 16.9 vs. 585.8, 34.7-fold). In B.1.351, B.1.617.2, and B.1.1.529, however, neutralizing antibody activity levels by booster injection tended to decrease as compared to WT (Booster injection_WT vs. B.1.351 vs. B.1.617.2 vs. B.1.1.529: 3537 vs. 1088 vs. 863.1 vs. 585.8). Interestingly, the neutralizing antibody activity ratio was maintained at 30–35-fold before and after booster injection (Figure 6). These results suggest that VOCs that evade neutralizing antibody activity can be counteracted by booster injections, but a targeted vaccine may be required as in B.1.1.529.

## 3. Discussion

Since vaccine-induced immune responses are antigen-specific responses, the level of antigenic protein delivered into the host, known as “vaccine immunogenicity”, is responsible for the success of candidate vaccines. Based on this strategy, to elicit potent and long-term immune responses against SARS-CoV-2, current various COVID-19 vaccine candidates exploit different platforms, including DNA-based [34], RNA-based [35,36], or viral vectored delivery systems [37,38].

In this study, we developed a novel COVID-19 vaccine platform, AdCLD-CoV19, a highly immunogenic, chimeric adenovirus-vectored vaccine that can stimulate antigen-specific humoral and cell-mediated immunity. To enhance the vaccine’s immunogenicity, we utilized two approaches: (1) inserting the linker peptide, which consists of repetitions of short amino acid sequences, into the furin cleavage site, and (2) switching the knob of adenovirus serotype 5 with type 35, which binds to CD46.

The linker peptide can confer structural stability to the antigenic protein, which leads to a more efficient expression of the protein product [39]. To evaluate whether candidate adenovirus-vectored vaccines can transduce various types of cells after intramuscular administration, we measured the level of expression of S-protein after viral transduction in THP-1(monocytes), A549 (epithelial cells), and RD (muscular cells) cell lines. Among our vaccine candidates, AdCLD-CoV19 induced high antigen expression in all tested cell lines, indicating that S-protein-encoding genes are stably overexpressed in various cell types in AdCLD-CoV19-vaccinated animals. Therefore, we suggest that the (GGGGS)1 linker, in addition to the approach of enhancing antigen overexpression by substituting particular amino acids in the S protein with proline (K986P, V987P) [28,29], could have a similar role.

Furthermore, the chimeric Ad5/35 can promote S-protein antigen delivery to lymphoid tissues and promote adaptive immunity as a result of improved transduction efficiency in CD46-expressing immune cells [40], especially resident antigen-presenting cells at the injection sites [41]. We demonstrated that AdCLD-CoV19 benefits from high levels of antigen expression and immunomodulation enabled by the Ad5/35 vector platform, leading to enhanced overall vaccine efficacy in mice and NHPs. According to our findings, a single dose of AdCLD-CoV19 significantly promoted the activity of high-level S protein-specific neutralizing antibodies and T cells mediated cytokines in mice and NHPs. This is a result of the complex immune response of AdCLD-CoV19, which can be explained as a result of an efficient vector platform that targets antigen-presenting cells as well as the high level of antigen productivity that can be obtained when the linker sequence is integrated into the antigen.

AdCLD-CoV19 requires a balanced evaluation of T cell activity as well as in terms of neutralizing antibody activity against vaccines due to the Th cell-biased response of the adenovirus itself [42]. Furthermore, there is growing evidence that suggests the COVID-19 vaccine strategy should engage T cell immunity in addition to neutralizing antibody activity [43,44,45]. We detected S-protein-specific CD8 T cells that can secrete more IFN-γ and TNF-α in vaccinated mice than in AdCLD-WT-treated mice. CD4^+^ T cell responses mediated by AdCLD-CoV19 were skewed toward the T_h_1 axis, as verified by cytokine analysis. The T_h_1/T_h_2 balance has been recognized as an important factor among vaccine researchers because vaccine-associated enhanced respiratory disease (VAERD) is largely associated with low-titer neutralizing antibodies and T_h_2-biased immune responses (related to IL-4 and IL-13 cytokines) [46,47]. Previous studies have reported that activating inflammatory antigen-presenting cells triggered by the adenovirus can promote T_h_1-skewed immune responses [19,40], which strengthens the idea that our Ad5/35 vector platform has been properly designed for safe and effective vaccines. Thus, our data suggest that AdCLD-CoV19 reduces the risk of VAERD by balancing the T_h_1-T_h_2 axis and achieving high titers of neutralizing antibodies.

Furthermore, in the cell-mediated immune response, studies have suggested that it is not antibodies but T cells that predominantly mediate long-term immunity to SARS-CoV-2 [48]. In the NHP challenge models, AdCLD-CoV19 reduced the viral load rather than spread it through proliferation and protected the tissue from infection of the respiratory tissue with live SARS-CoV-2, which is considered to be due to the T cell response. Therefore, we expect that AdCLD-CoV19 vaccination is a harmonious outcome of a staging T_h_1 response of inflammatory APCs recruited by adenoviral vectors and an antigen-specific humoral immune response following sufficient antigen expression, providing sufficient protection against COVID-19.

Even though AdCLD-CoV19 has adequate efficacy against COVID-19, the constantly arising SARS-CoV-2 variant complicates vaccination strategies. Variants of concern (VOC), defined by the US CDC, are the consequence of natural selection through which SARS-CoV-2 can optimize its entry to human cells by modifying unstable RNA genome. We hypothesized that the polyclonal antibody repertoire generated by AdCLD-CoV19 would be able to defend a wide spectrum of naturally occurring variants. In our data, the neutralizing antibodies in the serum of AdCLD-CoV19-vaccinated NHPs retained their neutralizing capacity against B.1.1.7, B.1.351, P.1, and B.1.617.2. However, we found several point-mutations require further attention, considering their varied sensitivity to vaccine-induced antibodies. The E484K mutation identified in B.1.351 and P.1 variants, for example, appeared to evade humoral immunity established by AdCLD-CoV19 vaccination, similar to previous studies [15,49]. In addition, the currently dominant variant, B.1.1.529 caused a change of the structure due to an extreme number of point mutations in the spike protein, which evaded the existing vaccine system and caused the breakthrough infection [12,50]. We tested whether the immune evasion of each mutation could be overcome with a booster shot. After a single administration of AdCLD-CoV19-1, a mass-production platform for AdCLD-CoV19, in which a booster shot was given at 9 weeks, significant neutralizing antibody activity was observed in VOCs including WT. However, B.1.1.529 shows relatively low neutralizing activity compared with other VOCs, suggesting that a B.1.1.529-only vaccine may be necessary from a long-term perspective.

Threatening point mutations can emerge at any time, assuming that the genome of SARS-CoV-2 will be ceaselessly diversified through natural selection. In the end, we estimate that the “whack-a-mole game” triggered by the increased incidence of new mutations will not end until next-generation vaccines are developed, or an effective heterogeneous prime-boost vaccine strategy is established. Therefore, further studies are needed to determine whether our vaccine can effectively counteract the various types of variants that may emerge in the future.

## 4. Materials and Methods

### 4.1. Cells and Adenovirus Design

Human embryonic kidney 293 cells expressing the tetracycline repressor (HEK293R), hACE2-expressing cell line (HEK293T-hACE2), A549 (ATCC, Manassas, VA, USA), and THP-1 (ATCC) cell lines were maintained in Dulbecco’s modified Eagle’s medium (Thermo Fisher Scientific, MA, USA) or RPMI 1640 medium (Thermo Fisher Scientific, Waltham, MA, USA) supplemented with 10% fetal bovine serum (Thermo Fisher Scientific, Waltham, MA, USA) and penicillin-streptomycin (100×, 5000 U/mL, Thermo Fisher Scientific, Waltham, MA, USA) at 37 °C in 5% CO_2_.

All replication-incompetent recombinant adenovirus vectored vaccines used in this study have deleted the E1 and E3 genes of adenovirus. The fiber, which is the cell receptor binding site of adenovirus serotype 5, was replaced with a knob of adenovirus serotype 35. AdCLD-S.WT is driven by the CMV promoter and was constructed using the SARS-CoV-2 codon-optimized spike protein gene, pCMV3-SARS-CoV-2 (GenBank ID: QHD43416.1) (Sino Biological, Beijing, China). The transmembrane and cytoplasmic domains of the spike protein are truncated for extracellular secretion. The furin cleavage site was mutated to the IL (AdCLD-S.IL), GGGGS (AdCLD-S.GS, AdCLD-CoV19), and (GGGGS)_3_ (AdCLD-S.GS3). The primer list is presented in Appendix A. The spike proteins modified in the adenovirus E1 deficient region with SwaI (New England Biolabs, Ipswich, MA, USA) were ligated through homologous recombination. All adenovirus-vectored products were purified by cesium chloride (Merck, Kenilworth, NJ, USA) density gradient ultra-centrifugation (1st step: 32,000 RPM for 90 min, 2nd step: 32,000 RPM for 18 h) and dialysis three times for 12 h with 2.5% glycerol-based buffer.

### 4.2. Western Blot Analysis

All vaccine candidates were transformed at 25 multiplicity of infection (MOI) into THP-1, A549, and RD cell lines seeded each 2.5 × 10^5^ cells. Western blot analysis was performed after harvested supernatant and lysate of cells transformed with vaccine candidates to verify the accumulation of S protein secretion. Following boiling in sample buffer without β-ME for detection of the S protein trimer, the protein samples were resolved on 4–12% Bis-tris gels. To prevent saturating the signal of spike antigen overexpression, each sample was exposed for less than 15 s.

Cells were lysed in RIPA buffer containing protease inhibitors (Merck, Kenilworth, NJ, USA), and the culture medium was concentrated using 50,000 MWCO microcon (Merck, Kenilworth, NJ, USA). Antibody-antigen complexes on PVDF membranes (iBlot 2 PVDF Mini Stacks, Thermo Fisher Scientific, Waltham, MA, USA) were quantified using Quantity One 1-D analysis software (Bio-Rad, Hercules, CA, USA). The following antibodies were used: rabbit anti-SARS-CoV-2 spike antibody (GeneTex, Irvine, MA, USA), mouse anti-β-actin antibody (Merck, Kenilworth, NJ, USA), goat anti-rabbit IgG H&L (HRP) (Abcam, Cambridge, MA, USA), and rabbit anti-mouse IgG (HRP) (Merck, Kenilworth, NJ, USA).

### 4.3. Animal Experiments

The animal experiments were approved and conducted following the guidelines of the International Animal Care and Use Committee (IACUC) of Seoul National University, Genia, and Korea Research Institute of Bioscience and Biotechnology (KRIBB).

For mouse immunization, 8-week-old BALB/c mice were injected intramuscularly once with 1 × 10^9^ VP of AdCLD-WT.S or AdCLD-CoV19 or 2 × 10^8^ VP of AdCLD-CoV19. Serum samples were collected from mice through retro-orbital bleeding at the indicated time points for each experiment. Mouse splenocytes were homogenized using a 70 μm cell strainer (BD Biosciences, San Jose, CA, USA), and red blood cells were removed with ACK lysis buffer (Thermo Fisher Scientific, Waltham, MA, USA) for further experiments.

For macaque immunization, 29–32-month-old cynomolgus macaques were immunized with 1 × 10^11^ VP of AdCLD-WT.S or AdCLD-CoV19, or 2 × 10^10^ VPs of AdCLD-CoV19. Serums and PBMCs were collected from immunized macaques at the indicated time points for each experiment. PBMCs were isolated by density gradient centrifugation using Histopaque^®^-1077 (Thermo Fisher Scientific, Waltham, MA, USA).

For the SARS-CoV-2 challenge in macaques, 6–7-year-old cynomolgus macaques (two males and four females) were injected intramuscularly with 2 × 10^10^ VP of Ad5/35-GFP or AdCLD-CoV19. To identify the immunogenicity of AdCLD-CoV19 in macaques, serums and PBMCs were collected before immunization and 3 weeks after immunization. On day 47 after immunization, macaques were challenged with a total of 10 mL of SARS-CoV-2 (2.6 × 10^6^ TCID_50_/mL, (BetaCoV/Korea/KCDC03/2020, accession no. 43,326 from the National Culture Collection for Pathogens)) via multiple routes (intratracheal (4 mL), oral (5 mL), intranasal (0.5 mL), and conjunctival (0.5 mL)) as previously described [33]. All macaques were anesthetized with ketamine sodium (10 mg/kg) and tiletamine/zolazepam (5 mg/kg) to measure body temperature and sample collection. Nasopharyngeal and oropharyngeal swab samples were collected before the challenge and 1, 2, and 3 days after the challenge. Swabs were centrifuged (1600× *g* for 10 min) and filtered with 0.2 µm pore size syringe filters for virus quantification. Macaques were euthanized 3 days after the challenge, and lung tissues were harvested. Lung samples were homogenized and centrifuged. The supernatants obtained after centrifugation were used for virus quantification.

### 4.4. ELISA and Immunospot Analysis

In the enzyme-linked immunosorbent assay (ELISA), ELISA plates (Thermo Fisher Scientific, Waltham, MA, USA) were coated with 100 ng/well recombinant spike protein (AcroBiosystems, Newark, DE, USA) in PBSN (PBS 1 L + sodium azide 0.01 g) at 4 °C for 16 h. Plates were washed three times with PBS and blocked with 150 µL blocking buffer (PBSN + BSA 1%) at 37 °C for 90 min. After washes, plates were incubated with serial dilution of serum samples for 3 h at 37 °C. Following washes, plates were incubated with the 1000-fold dilution of Goat Anti-mouse Ig-HRP (BD Biosciences, CA, USA) or Goat anti-human Ig-HRP (SouthernBiotech, Birmingham, AL, USA) for 2 h at 37 °C. After final washes, plates were developed by TMB reagent (KPL, Milford, MA, USA) and were stopped using 0.25 N HCl. Samples were analyzed at a wavelength of 450 nm using a microreader. Endpoint titers were calculated as the dilution factor that emitted an OD value (450 nm) exceeding 3× OD value of wells that were incubated with secondary antibody alone.

IFN-γ ELISpot analysis was performed using mouse splenocytes and PBMCs or non-human primate PBMCs. The pre-coated plate was incubated at 37 °C for 20 min with 4 pools of 8 amino acids-overlapped 15-mer peptides from SARS-CoV-2 spike protein (Genscript, Piscataway, NJ, USA) at a concentration of 1µg/mL per peptide. The details of peptide pools are summarized in Appendix A. Thereafter, mouse splenocytes, PBMCs, and non-human primate PBMCs were added to each well and cultured at 37 °C for 20 h. 1 μg/mL anti-CD3 mAb (BioLegend, San Diego, CA, USA) or 50 ng/mL PMA/500 ng/mL ionomycin (Merck, Kenilworth, NJ, USA) were used as positive controls. For the negative control, only media (RPMI 1640 supplemented with 10% FBS) was used. IFN-γ-specific ELISpot development in mice and non-human primates was performed using the Mouse IFN-γ ELISpot kit (CTL Europe GmbH, Bonn, Germany) or Human IFN-γ ELISpot kit (CTL Europe GmbH, Bonn, Germany). Spots on the dried plate were counted using Immunospot S6 microanalyzer (CTL Europe GmbH, Bonn, Germany). Spot-forming unit (SFU) 1 × 10^6^ PBMCs were calculated by subtracting the negative control wells and were summed across the 4 peptide pools.

### 4.5. Intracellular Cytokine Staining

The antibodies used for flow cytometry were as follows: anti-mouse CD3ε (145-2C11), anti-CD4 (GK1.5), anti-CD8α (53-6.7), anti-CD44 (IM7), anti-IFN-γ (XMG1.2), anti-TNF-α (MP6-XT22), anti-IL-4 (11B11), and anti-IL-5 (TRFK5). All Abs were purchased from BioLegend (CA, USA). Fixable Viability Dye (eBioscience, San Diego, CA, USA) was used to exclude dead cells from the analysis.

For T cell intracellular staining analysis, cells were restimulated with peptide pools of SARS-CoV-2 spike protein at a concentration of 0.5µg/mL per peptide (Genscript, Piscataway, NJ, USA) in the presence of 1µg/mL BD GolgiPlug (BD Biosciences, San Jose, CA, USA) for 6 h. Single-cell suspensions were stained for surface markers for 30 min at 4 °C. Cells were fixed and permeabilized using a Cytofix/Cytoperm kit (BD Biosciences, San Jose, CA, USA) according to the manufacturer’s protocol. Stained cells were washed with FACS buffer (0.2% BSA, 0.1% sodium azide, and 2 mM EDTA). Samples were acquired on a LSRFORTESSA (BD Biosciences, San Jose, CA, USA) and analyzed using FlowJo (Tree Star, Ashland, OR, USA).

### 4.6. In Vivo Cytotoxicity Assay

Spike protein-specific cytolytic activity of CD8 T cell responses in vaccinated mice was measured using flow cytometry. BALB/c mice were vaccinated with 1 × 10^9^ VP of AdCLD-CoV19. Two weeks after vaccination, splenocytes were obtained from syngeneic mice and lysed with ACK lysis buffer. Cells were either loaded with peptide pools at a concentration of 0.5 µg/mL per peptide or left untouched and labeled with CFSE at different concentrations (5 or 0.5 μM, respectively). Ag-pulsed (CFSE^hi^) and unpulsed (CFSE^low^) cells were mixed equally and injected intravenously into AdCLD-CoV19-vaccinated or naïve control mice. Recipient mice were sacrificed at 20 h after injection. CFSE-labeled cells in splenocytes were analyzed to determine Ag-specific cytolytic activity. Spike-specific lysis was calculated as follows: percent (%) lysis = [1 − {(CFSE^low^_control/_CFSE^hi^_control_)/{(CFSE^low^_vaccinated/_CFSE^hi^_vaccinated_)}] × 100.

### 4.7. Pseudotyped Lentivirus and Neutralization Assay

Neutralization against pseudotyped lentivirus expressing SARS-CoV-2 S protein was performed using a luciferase assay. The lentivirus VSVG gene was replaced with the plasmid expressing SARS-CoV-2 S protein, and the lentiviral vector backbone carries the reporter genes for firefly luciferase and GFP. The pseudovirus was produced by co-transducing the plasmid expressing SARS-CoV-2 S protein, the lentiviral vector backbone, and the lentiviral vector packaging plasmid into HEK293T cells.

The sera from vaccinated mice or non-human primates were diluted four times in a row from 25-fold to 6400-fold and incubated with pseudotyped lentivirus for 1 h at 37 °C. After 1 h, the mixture was added to HEK293T-hACE2 cells that express human ACE2. The HEK293T-hACE2 cell line was produced from HEK293T cells by transducing lentiviral vector harboring the hACE2 gene and then puromycin (Thermo Fisher Scientific, Waltham, MA, USA) single-cell sorting (Appendix A). Polybrene was added to the cells and the mixture at a final concentration of 10 µg/mL. At 48 h after the infection, luciferase activity in the cell lysates was measured using a luciferase assay kit (Promega, Madison, WI, USA) and a luminometer (Centro LB960; Berthold Technologies, Baden-Württemberg, Germany). The neutralization abilities of the sera were calculated as IC50, which is the dilution factor in which the luciferase activity was reduced to 50% of that from the virus-only wells.

### 4.8. Live Virus Neutralizing Test (PRNT)

Vero cells were seeded in 6 well plate and incubated at 37 °C and 5% CO_2_ for 16–18 h. Serum samples at 6 weeks after the first immunization were inactivated at 56 °C for 30 min and 2-fold serially diluted from 20-fold diluted samples. Diluted serum samples were mixed with SARS-CoV-2 (50–100 plaque-forming unit/well) at 37 °C and 5% CO_2_ for 1 h and then inoculated into Vero cells for 1 h under the same incubation conditions. After removal of the serum-virus mixture, 1.5% low melting temperature agarose in DMEM supplemented with 2% fetal bovine serum was added and incubated for 3 days. The plates were fixed with 10% neutral formalin solution overnight and stained with 0.1% crystal violet solution. For each dilution, the number of plaques was counted, and the percentage of neutralization was compared with the control.

### 4.9. Virus Identification and Quantification

To calculate the values of TCID_50_/mL, all tissue and swab samples from infected macaques were diluted and inoculated into Vero cells and incubated for 3 days at 37 °C and TCID_50_/mL values were determined using the Reed and Muench method.

To quantify the SARS-CoV-2 virus, the viral RNA genome was extracted from the samples using the QIAamp Viral RNA Mini Kit (Qiagen, Hilden, Germany). Quantitative reverse transcription-polymerase chain reaction (qRT-PCR) was performed with a primer set targeting partial regions of the ORF1b gene in SARS-CoV-2 using the QIAGEN OneStep RT-PCR Kit as reported previously [51]. SARS-CoV-2 RNA standard and negative samples were run in parallel in all qRT-PCR analyses to calculate the virus copy number.

### 4.10. Statistical Analysis

Comparisons between two groups were made using the two-tailed Student’s *t*-test and one-way analysis of variance (ANOVA). Statistical significance was set at *p* < 0.05. GraphPad Prism 5 version 5.03 software (GraphPad, San Diego, CA, USA) was used for the analyses.

## Figures and Tables

**Figure 1 vaccines-10-00712-f001:**
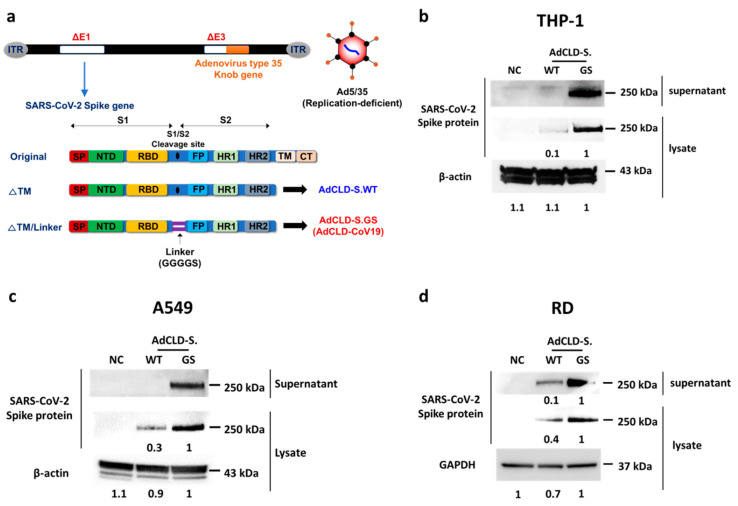
Adenovirus-vectored vaccine construction and antigen characterization. (**a**) Schematic diagram of the structure of adenovirus vectored vaccine harbored with modified SARS-CoV-2 S protein trimer. (**b**–**d**) AdCLD-S.WT or AdCLD-CoV19 was transformed into THP-1, A549, and RD cell lines (each 2.5 × 10^5^ cells) at each 25 multiplicity of infection (MOI). The accumulation level of S protein secretion was verified by Western blot analysis after harvesting supernatant (40 µg/well) and cell lysate (40 µg/well) of cells transformed with AdCLD-S.WT or AdCLD-CoV19. The protein samples were resolved on 4–12% Bis-tris gel after being boiled in sample buffer without β-ME for detection of the S protein trimer. Each sample was exposed for less than 15 s to prevent saturating the signal of spike antigen overexpression. SP, signal sequence; NTD, N-terminal domain; RBD, receptor-binding domain; FP, fusion peptide; HR1, heptad repeat 1; HR2, heptad repeat 2; TM, transmembrane domain; CT, cytoplasmic tail; NC, Negative control.

**Figure 2 vaccines-10-00712-f002:**
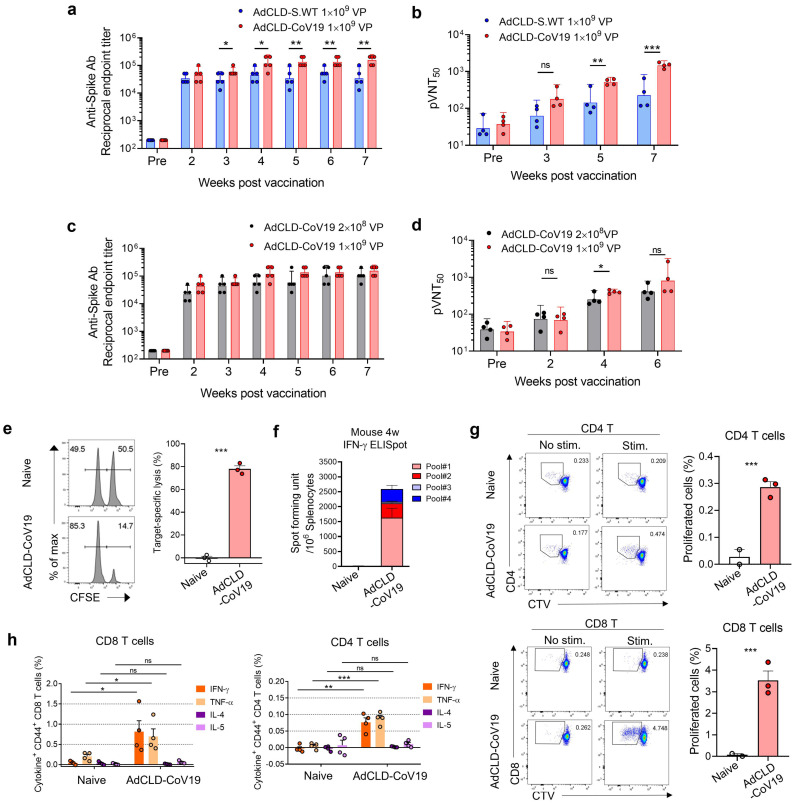
Immunogenicity of AdCLD-CoV19 in mice. (**a**,**b**) Naïve BALB/c mice were vaccinated with 1 × 10^9^ VP of AdCLD-CoV19 or AdCLD-S.WT with a single intramuscular injection and serums were collected at indicated time points. Serums were assessed for S-specific Abs by ELISA (**a**) and neutralizing Abs by SARS-CoV-2 pseudovirus neutralization test, pVNT50, 50% pseudovirus neutralization titer (**b**). (**c**,**d**) Naïve BALB/c mice were vaccinated with 1 × 10^9^ VP or 2 × 10^8^ VP of AdCLD-CoV19, and collected serums were assessed for S-specific Abs by ELISA (**c**) and neutralizing Abs by SARS-CoV-2 pseudovirus neutralization test (**d**). (**e**) In vivo cytotoxicity analysis of AdCLD-CoV19-vaccinated mice. CFSE^hi^, peptide-loaded target; CFSE^lo^, peptides-unloaded control. The percentages of specific-peptides-loaded target cell lysis are shown as a graph. (**f**) IFN-γ-secreting T cells were measured by ELISpot assay from splenocytes stimulated with each peptide pool (see Methods). (**g**) CD4 (upper) and CD8 (lower) T cell proliferation under peptides stimulation was assessed by flow cytometry. The percentages of T cells that proliferated at least once are shown as graphs. (**h**) Ten weeks after vaccination, splenocytes were restimulated with peptide pools for 6 h and intracellular cytokine staining (ICS) was conducted to measure T cell responses. The percentages of cytokine expression from CD4 (left) and CD8 (right) T cells are depicted as graphs. Two-tailed Student’s *t*-test was used for the analysis. *, *p* < 0.05; **, *p* < 0.01; ***, *p* < 0.001; ns, not significant.

**Figure 3 vaccines-10-00712-f003:**
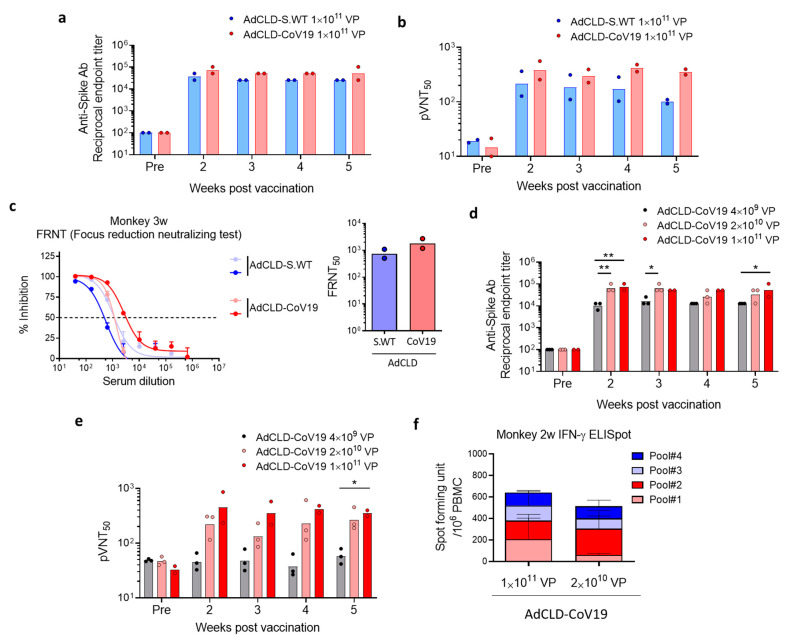
Immunogenicity of AdCLD-CoV19 in macaques. (**a**–**c**) Naïve cynomolgus macaques received 1 × 10^11^ VP of AdCLD-CoV19 or AdCLD-S.WT with a single intramuscular injection and serums were collected at indicated time points. Serums were assessed for S-specific Abs by ELISA (**a**) and neutralizing Abs by SARS-CoV-2 pseudovirus neutralization test, pVNT50, 50% pseudovirus neutralization titer (**b**). (**c**) Focus-reduction neutralizing tests (FRNT) were conducted using serum samples collected 3 weeks post-vaccination. (**d**–**f**) Cynomolgus macaques were vaccinated with 1 × 10^11^ VP or 2 × 10^10^ VP of AdCLD-CoV19, and their immunogenicity was accessed at the indicated time points. (**d**) S-specific Ab ELISA, (**e**) pseudovirus neutralization assay, and (**f**) IFN-γ ELISpot assay. Two-tailed Student’s *t*-test (**a**–**c**,**e**) and one-way analysis of variance with Tukey’s multiple comparison (ANOVA) (**d**,**e**) were used for the analysis. *, *p* < 0.05; **, *p* < 0.01.

**Figure 4 vaccines-10-00712-f004:**
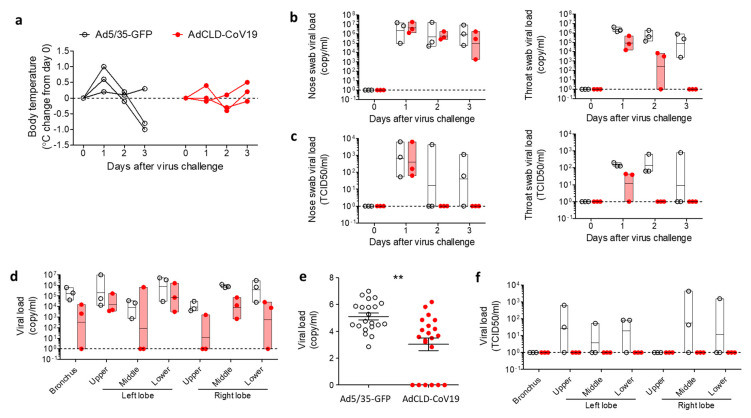
Protective efficacy of AdCLD-CoV19 against SARS-CoV-2 challenge. AdCLD-CoV19 vaccinated cynomolgus macaques and control vector (Ad5/35-GFP) administered macaques were challenged with SARS-CoV-2 at 7 weeks post-vaccination. (**a**) Body temperature changes in macaques after the challenge are shown in a graph. (**b**) Viral RNA from the nose (left) and throat (right) swab samples of macaques at the indicated time points were measured by qRT-PCR. (**c**) Infectious viral loads of the nose (left) and throat (right) swab samples were measured using the TCID50 assay. (**d**) Lung tissues were collected from macaques on day 3 post-challenge and quantified for viral RNA using qRT-PCR. (**e**) The pooled data of all examined lung tissues are shown as a graph. (**f**) Infectious viral loads in lung tissues were measured using the TCID_50_ assay. Two-tailed Student’s *t*-test was used for the analysis. **, *p* < 0.01.

**Figure 5 vaccines-10-00712-f005:**
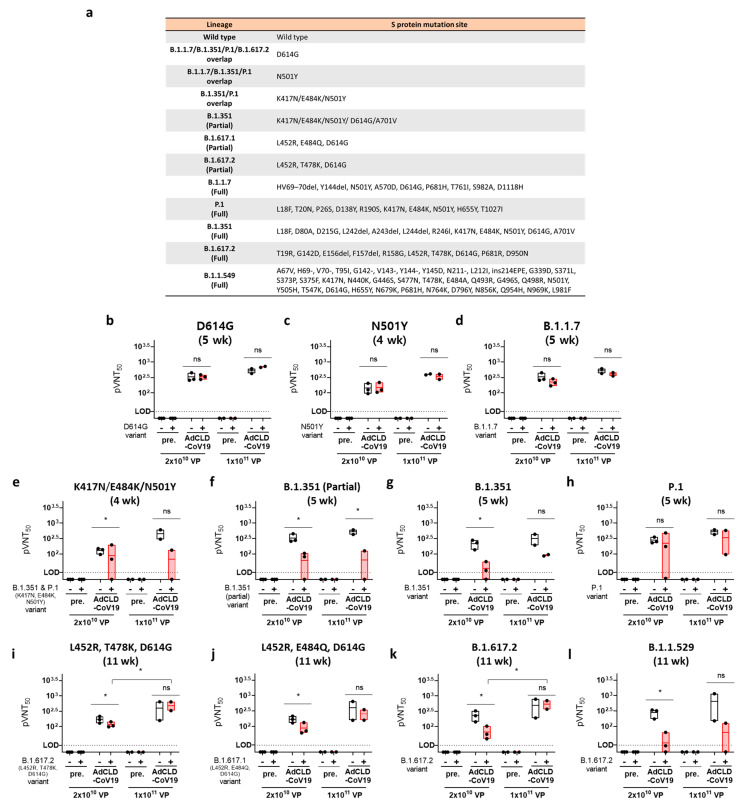
Evaluation of neutralization titer against SARS-CoV-2 variants of AdCLD-CoV19. (**a**) The pseudotyped lentivirus list table. (**b**–**l**) Sera from NHPs immunized for 4, 5, or 11 weeks with AdCLD-CoV19 (2 × 10^10^ or 1 × 10^11^ VP) were incubated with each variant pseudotyped lentivirus (1 × 10^6^ TU/mL) for 1 h. The mixture was reacted with the HEK293T-hACE2 cell line (1 × 10^4^ cells/well in 96-well plates), expressing luciferase for 2 days. The ‘−’ symbol in the graphs indicates a pseudovirus reflecting the Wuhan-Hu-1 wild-type, and a ‘+’ symbol indicates a pseudovirus reflecting each mutation. The reciprocal neutralizing titers on the pseudotyped lentivirus neutralization test at a 50 % pseudovirus neutralization titers (pVNT_50_) are shown. All graphs are displayed with floating bar (min to max) boxes and mean values indicated by inner lines, with horizontal dashed lines representing the limit of detection (LOD). Two-tailed Student’s *t*-test was used for the analysis. *, *p* < 0.05; ns, not significant.

**Figure 6 vaccines-10-00712-f006:**
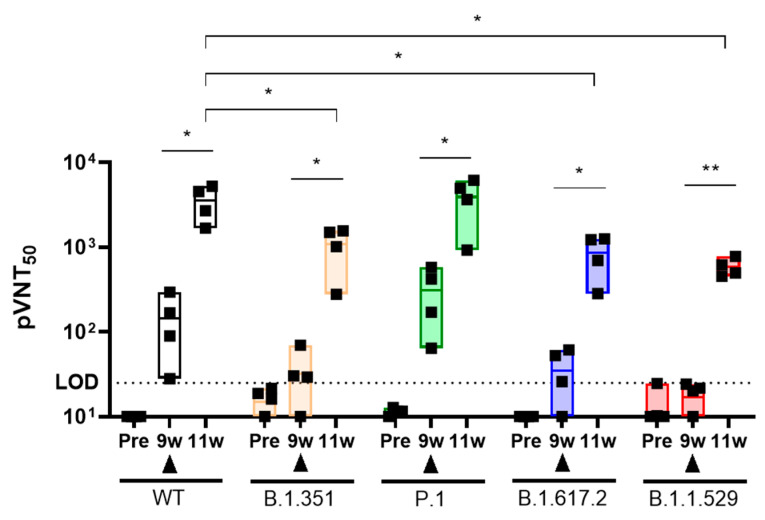
Booster shot evaluation of AdCLD-CoV19-1. NHPs were immunized by AdCLD-CoV19-1 (2 × 10^10^ VP) once intramuscularly, followed by booster injections at the same dose at 9 weeks. Immunized sera from weeks 9 and 11 responded for 1 h with each variant (WT, B.1.351, P.1, B.1.617.2, B.1.1.529) pseudovirus (1 × 10^6^ TU/mL). The neutralizing antibodies titer from the mixture was determined by incubating the combination for 2 days with the luciferase-expressing HEK293T-hACE2 cell line (1 × 10^4^ cells/well in 96-well plates). The reciprocal neutralizing titers on the pseudotyped lentivirus neutralization test at pVNT50 are shown. All graphs are displayed with floating bar (min to max) boxes and mean values indicated by inner lines, with horizontal dashed lines representing the limit of detection (LOD). The closed triangle means booster injection. Two-tailed Student’s *t*-test was used for the analysis. *, *p* < 0.05; **, *p* < 0.01.

## Data Availability

All data is included in the manuscript and its Appendix A, or can be obtained by contacting the corresponding author.

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
