# Peer review of "The Chimeric Adenovirus (Ad5/35) Expressing Engineered Spike Protein Confers Immunity against SARS-CoV-2 in Mice and Non-Human Primates"

_vaccines, 2022, doi:10.3390/vaccines10050712_

Round 1
Reviewer 1 Report
This paper reports the findings from COVID-19 immunity vaccines research. Overall, the research appears to follow standard research protocols and these are reported relatively well in the paper.
The one thing that is inadequate in the text of the paper is the description and presentation of the statistical analyses. The text of the paper needs to include tables that carefully report all the relevant statistical data for the mice and non-human primates research and the associated statistical tests of significance for the findings. The graphs presented in the paper are good and helpful, but the specifics of the statistical analysis comparisons and associated tests of significance need to be included.
Author Response
Reviewer 1: Comments and Suggestions for Authors
This paper reports the findings from COVID-19 immunity vaccines research. Overall, the research appears to follow standard research protocols and these are reported relatively well in the paper.
The one thing that is inadequate in the text of the paper is the description and presentation of the statistical analyses. The text of the paper needs to include tables that carefully report all the relevant statistical data for the mice and non-human primates research and the associated statistical tests of significance for the findings. The graphs presented in the paper are good and helpful, but the specifics of the statistical analysis comparisons and associated tests of significance need to be included.
Answer:
- We appreciate the comments by reviewer 1. We used two-tailed student’s t-test for comparing 2 groups and one-way ANOVA with Tukey's multiple comparisons for comparing more than 3 groups. We have added the details of statistical analysis in each figure legend.
In addition, according to the reviewer’s suggestion, we have added statistical summary tables of Ab responses induced by AdCLD-CoV19. Please find Supplementary Table 1 and 2. We hope that this information would be helpful to readers.
Reviewer 2 Report
Possible corrections:
- Typing mistakes
p.2. References 14, 15 in Superscript
the receptor on host cells, through the receptor-binding domain (RBD)14,15
p.2. missed accession numbers
Representatively the Delta (B.1.617.2) or Omicron (B.1.1.529)
p.2. missed space before brackets
ACE2(angiotensin-converting enzyme 2)
- Different variants within the text:
Th1 on p1and Th1 on p.2.
BALB/c on p.5 and Balb/C on p.7
- Figures within the text and supplemental figures are the same, but author refer to either Figure 3, p.8 either to Supplementary Figure 3, p.9.
- Paragraf on p. 11. is written in different letter size.
The neutralizing antibody titer for the B.1.351 variant, on the other hand, was significantly reduced than the wild type (GMT: M_WT vs. Variant: 216 vs. 33.7, 6.4-fold; H_WT vs. Variant: 315.6 vs. 90.8, 3.5-fold), and the key mutations of B.1.351 (partial) and K417N/E484K/N501Y followed a similar tendency (Figure 5e-g). Furthermore, the neutralizing antibody activity against B.1.1.529, which also contains the 417/484/501 site point mutation, decreased in a similar pattern to that of B.1.351 (GMT: M_WT vs. Variant: 279.6 vs. 29.4, 9.5-fold; H_WT vs. Variant: 650.7 vs. 63.7, 10.2- fold) (Figure 5l). Interestingly, the P.1 variant, which shares important mutations with B.1.351 and B.1.1.529, restored neutralizing activity compared to B.1.351 (GMT: M_WT vs. Variant: 278.5 vs. 225.2, 1.2-fold; H_WT vs. Variant: 521.1 vs. 338.9, 1.5-fold) (Figure 5h).
- In the section Materials and methods
a) not for all materials are given necessary data (XXX Company, City, Country);
b) for cesium chloride density gradient ultra-centrifugation and Western blot: methods shoul be briefly described or appropriately cited.
c) I didn’t find Table S1 p.17 and Table XX p.19.
Author Response
Reviewer 2: Comments and Suggestions for Authors
Possible corrections:
- Typing mistakes
- p.2. References 14, 15 in Superscript
- p.2. missed accession numbers
à Representatively the Delta (B.1.617.2) or Omicron (B.1.1.529)
- p.2. missed space before brackets
à ACE2(angiotensin-converting enzyme 2)
- Different variants within the text:
àTh1 on p1and Th1 on p.2.
à BALB/c on p.5 and Balb/C on p.7
- Figures within the text and supplemental figures are the same, but author refer to either Figure 3, p.8 either to Supplementary Figure 3, p.9.
- Paragraf on p. 11. is written in different letter size.
à The neutralizing antibody titer for the B.1.351 variant, on the other hand, was significantly reduced than the wild type (GMT: M_WT vs. Variant: 216 vs. 33.7, 6.4-fold; H_WT vs. Variant: 315.6 vs. 90.8, 3.5-fold), and the key mutations of B.1.351 (partial) and K417N/E484K/N501Y followed a similar tendency (Figure 5e-g). Furthermore, the neutralizing antibody activity against B.1.1.529, which also contains the 417/484/501 site point mutation, decreased in a similar pattern to that of B.1.351 (GMT: M_WT vs. Variant: 279.6 vs. 29.4, 9.5-fold; H_WT vs. Variant: 650.7 vs. 63.7, 10.2- fold) (Figure 5l). Interestingly, the P.1 variant, which shares important mutations with B.1.351 and B.1.1.529, restored neutralizing activity compared to B.1.351 (GMT: M_WT vs. Variant: 278.5 vs. 225.2, 1.2-fold; H_WT vs. Variant: 521.1 vs. 338.9, 1.5-fold) (Figure 5h).
- In the section Materials and methods
- a) not for all materials are given necessary data (XXX Company, City, Country);
- b) for cesium chloride density gradient ultra-centrifugation and Western blot: methods shoul be briefly described or appropriately cited.
- c) I didn’t find Table S1 p.17 and Table XX p.19.
Answer:
- Reviewer 2 adequately reviewed the details we were missing. We accept most of Reviewer 2's revision suggestions.
- Reviewer 2 mentioned '1. typing mistakes' and '2. different variants within the text,' and '4. different letter size' in the text has been suitably repaired.
-> Page 2: References 14 and 15 changed the superscript style to the vaccines style.
-> Page 2: the Delta (B.1.617.2) or Omicron (B.1.1.529) to the Delta (B.1.617.2, accession number: MZ208926.1) or Omicron (B.1.1.529, accession number: OL869974.1)
-> Page 2: ACE2(angiotensin-converting enzyme 2) à ACE2 (angiotensin-converting enzyme 2)
-> Page 2 or 5~6: We have unified the terminology 'Th1' and 'BALB/c'.
-> Page 10: We changed the size 11 paragraphs on page 11 of the pre-edited file to size 10.
- The revision requests '3' and '5c' are not our intention, and it seems that the actual 'supplemental figures' were not reflected in the review of the paper. We have requested the editorial staff to review the missing supplemental figures.
- The revision requests '5a' and '5b' added more information to the text to ensure that the reader has enough information.
-> Page 16: We have provided a detailed description of the virus purification method using cesium chloride and Western blot analysis in the 'Methods and Materials' section.
Please see the attachment.

Round 2
Reviewer 1 Report
The revisions to this paper have been responsive to the reviews of the previous version, and the manuscript is improved.